# Menopausal Symptoms and Perimenopausal Healthcare-Seeking Behavior in Women Aged 40–60 Years: A Community-Based Cross-Sectional Survey in Shanghai, China

**DOI:** 10.3390/ijerph17082640

**Published:** 2020-04-12

**Authors:** Li Du, Biao Xu, Cheng Huang, Liping Zhu, Na He

**Affiliations:** 1Department of Epidemiology, School of Public Health, Fudan University, The Key Laboratory of Public Health Safety of Ministry of Education, Shanghai 200032, China; lilydu82@126.com (L.D.); bxu@shmu.edu.cn (B.X.); 16211020002@fudan.edu.cn (C.H.); 2Department of Research and Education, Shanghai Center for Women and Children’s Health, Shanghai 200062, China; 3Key Laboratory of Health Technology Assessment of Ministry of Health, Fudan University, Shanghai 200032, China

**Keywords:** menopausal symptoms, perimenopausal healthcare, associated factors

## Abstract

The aim of the study was to specify prevalence and severity of menopausal symptoms among middle-aged women and to understand the factors associated with women’s perimenopausal healthcare-seeking behavior in Shanghai, China. A community-based cross-sectional study was carried out involving 3147 participants aged 40–60 years. A combination of stratified sampling and quota sampling was used. Out of the total 16 districts in Shanghai, 7 were purposefully selected in consideration of covering both central and suburban areas, population distribution, and willingness to participate. Two communities were randomly selected in each of six districts. Four communities were randomly selected in the 7th district considering the relatively low coverage of central population in the sampling frame. Eligible women were recruited continuously according to the house number and invited to participate in the study until 200 participants were recruited in each community. A structured questionnaire was designed to collect information including sociodemographic data, menopausal symptoms, and experiences in seeking perimenopausal healthcare. The severity of menopausal symptoms was assessed with the modified Kupperman menopausal index (mKMI). The mean age of all the participants was 51 years. 33.13% of the participants were premenopausal, 14.52% were perimenopausal, and 52.35% were postmenopausal. The total prevalence of menopausal symptoms was 73.8%, while among the perimenopausal women, the symptoms were the most common (81.70%). The top three reported symptoms were fatigue (38.08%), hot flushes and sweating (33.65%), and joint ache (28.81%). Perimenopausal and postmenopausal participants had a higher score of the mKMI than premenopausal women (*p* < 0.01). Of the women who had symptoms, 25.97% had sought healthcare. A logistic regression model revealed that employment, menstruation status, and the mKMI were significantly associated with healthcare-seeking behaviors (*p* < 0.01). We concluded that prevalence of menopausal symptoms was relatively high among middle-aged women, with perimenopausal women showing the highest level. However, only a small percentage of the participants sought healthcare. Carrying out health education may be a measure to improve the healthcare-seeking behavior.

## 1. Introduction

In most countries, there is an increase in the aging population as a result of both longer life expectancy and declining fertility rates [1]. The World Health Organization has adopted a Global strategy and action plan on ageing and health to ensure adults live not only longer but healthier lives. Demographic data have shown that every year, 25 million women worldwide experience the menopause. This will result in 1.2 billion postmenopausal women worldwide by 2030 [2]. The menopause in white Caucasian women occurs on average at age 51 (with ethnic and regional variations) [3], while in Asian countries, the mean age of onset of the menopause varies, with the mean age of onset in Taiwanese women at 49.3 years as reported in a study in 1997 [4].

During the menopausal period, the cessation of the ovarian hormonal function, especially the limited level of estrogens, may lead to the development of vasomotor, psychological, somatic, and atrophic changes in the estrogen-dependent tissues, which contribute to the so called menopausal (climacteric) syndrome. The physiological and pathological alterations may exert a negative effect on women’s quality of life, and even cause severe physical and mental illness [5].

As women complete the transition to the menopause, an estimated 85% of women report at least one menopausal symptom, and only 10% of these women would seek healthcare [6]. There is evidence of differences in prevalence and composition of menopausal symptoms in Asian women comparing to Western Caucasian women. In Europe and North America, the most common symptoms reported by women during the menopause transition are hot flushes and night sweats, which affect approximately 70% of women. According to the Hilditch’s study, Chinese women from Guangzhou reported a lower frequency of symptoms and experienced less distress from symptoms than Canadian women [7]. An investigation conducted in Hong Kong showed that, compared to pre- and postmenopausal women, perimenopausal women had the largest number of reports of symptom complaints. Musculoskeletal conditions constituted the top complaints reported by the respondents, followed by headaches and psychological symptoms [8]. In two reports of vasomotor symptoms in Asian women, more Chinese women suffered from hot flushes and night sweats than Thai women [9,10].

In China, in 2015, the female population of 40–60-year-olds was about 322 million, with a life expectancy of 79.43 years [11]. It is important to understand the prevalence of perimenopausal symptoms in Chinese women to identify the needs of healthcare in order to improve life quality of middle aged and elderly women. To date, there are very few studies investigating prevalence of menopausal symptoms among women on the community population level and their healthcare-seeking behaviors. Therefore, we designed the present study to understand the situation with menopausal symptoms and relevant healthcare-seeking behaviors among perimenopausal women aged 40–60 years in Shanghai communities. The findings shall provide evidence for appropriate health policy-making for menopausal care.

## 2. Materials and Methods

### 2.1. Study Design and Sampling

A community-based cross-sectional study was carried out, using a combination of stratified sampling and quota sampling. Out of the total 16 districts in Shanghai, 7 were purposefully selected, in consideration of covering both central and suburban areas, population distribution, and willingness to participate. Two communities were randomly selected in each of six districts. In the seventh district, which is located in central Shanghai, 4 communities were randomly selected considering the relatively low coverage of central population in the sampling frame. As a total, 16 communities were included. Eligible women were approached continuously according to the house number and invited to participate in the study until 200 participants were recruited in each community. The study participants were women aged 40 to 60 years. Informed consent was received from each participant. Face-to-face interviews by trained health workers were conducted to support each woman’s filling of a self-administered questionnaire. Women with any of the following conditions were excluded from the study: non-Shanghai registered residents; a history of bilateral oophorectomy or hysterectomy; very severe psychological disease; low intelligence.

### 2.2. Data Collection

Information of the participants, including demographic characteristics, socioeconomic status, family, medical history, perimenopausal symptoms, health demands, and health-related behaviors, were collected.

According to the STRAW+10 (Stages of Reproductive Aging Workshop+10) definition, the menopausal stage is defined as: premenopause, regular menstrual cycles within the past year; perimenopause, irregular menstrual cycles within the past year; postmenopause, no menstruation within the past year. Serum levels of estradiol (E_2_) and follicle-stimulating hormone (FSH) were measured as indicators of the ovarian function [12]. In the perimenopausal stage, the early folicular phase (FP) FSH levels increase, the mean estradiol or estrogen excretion levels are not low. The perimenopause is characterized by higher average (compared with premenopausal women) and erratic estrogen levels. Paradoxically, the perimenopause is associated with a significant rate of spinal bone mineral density (BMD) loss. In the postmenopausal stage, FSH continues to increase and estradiol continues to decrease until approximately 2 years after the final menstrual period, after which the levels of each of these hormones stabilize [13].

The modified Kupperman menopausal index was used to score the menopausal symptoms of the participants which had been commonly observed in China and verified to be more suitable to Chinese women [14]. In addition to the 11 items included in the original Kupperman Index (KI), the modified version adds two urogenital symptoms, urinary infection and sexual complaints. The index was calculated according to the replies concerning intensity of the following complaints: (1) sweating and hot flushes, (2) paresthesia, (3) insomnia, (4) nervousness, (5) melancholia, (6) vertigo, (7) fatigue, (8) arthralgia/myalgia, (9) headache, (10) heart palpitation, (11) formication, (12) sexual complaints, and (13) urinary tract infection. Each item was rated with a 4-point Likert scale and was scored as follows: not present—0, mild—1, moderate—2, severely expressed—3. According to the total score, women were qualified into four groups: no obvious menopausal syndrome—less than 6 points, mild symptoms—7–15 points, moderate symptoms—16–30 points, and severe symptoms—more than 30 points.

### 2.3. Statistical Analysis

The data from all questionnaires and interviews were included in the statistical analysis using the Epidata 3.1 software (EpiData Association, Odense, Denmark). SPSS 21.0 for Windows (IBM Corp., Armonk, NY, USA) was used to analyze the data. A descriptive analysis was used to describe characteristics of the participants and their healthcare-seeking behaviors. The chi-squared test was performed to compare different symptoms between three menopausal stages of the participants, and the factors associated with healthcare-seeking behaviors. The factors which were significantly related to healthcare seeking were then included in the logistic model analysis. A multivariable logistic regression model was built to identify the determinants of healthcare-seeking behaviors.

### 2.4. Ethical Considerations

The Ethics Committee of the Fudan University approved the study (IRB # 2016-03-0579).

## 3. Results

### 3.1. Characteristics of the Study Population

Of the 3219 women interviewed, 72 were excluded because of too much missing information in the questionnaires; the response rate was 97.8%. Socioeconomic and health characteristics of the participants are presented in Table 1. For the total of 3147 participants, the average age was 51 years (standard deviation (SD): 5.80), and 52.35% of them had the menopause. The majority of the participants (73.98%) were well-educated, with the senior high school level or higher. 97.12% of the women had health insurance. Among the participants, 14.52% were at the perimenopausal stage, 52.35% were at the postmenopausal stage.

### 3.2. Symptoms of the Study Population at Different Menopausal Stages

The total prevalence of menopausal symptoms was 73.80%. Fatigue, hot flushes/sweating, and joint ache were the most common symptoms, reported by 38.08%, 33.65%, and 28.81% of the women, respectively. The prevalence of most symptoms, such as hot flushes/sweating, fatigue, and emotional disorder among the participants in the perimenopausal period was significantly higher than in the women in the pre- and post-menopausal periods: 43.18%, 44.52%, and 21.25% respectively (*p* < 0.01). During the postmenopausal stage, hot flushes/sweating, fatigue, and joint ache were most common symptoms, reported by 42.12%, 34.43%, and 33.81% of the women, respectively. The prevalence of height decrease and bone fracture was higher at the postmenopausal stage than at other stages, which were 8.93% and 3.04% (Table 2).

A total of 2943 participants completed the Kupperman menopausal index (KMI) scale, with the response rate of 94.09%. Almost half (48.52%) of the women scored less than 6. The women in the premenopausal period had the lowest KMI score, while the perimenopausal and postmenopausal participants had higher KMI scores. In total, 33.74% women had mild symptoms, 16.55%—moderate, and 1.19%—severe symptoms.

### 3.3. Healthcare-Seeking Experience for Menopausal Complaints

There were 602 participants seeking healthcare services, accounting for 25.97% of all the women with menopausal symptoms. Only 8.49% of them went to the menopausal department; the majority of women went to seek care at the department of internal medicine (39.43%), and 22.17% sought care at the gynecology department. Only 10.54% had estrogen tested and only 4.01% were given the perimenopausal scale test. The majority took the general blood test and the electrocardiogram (ECG) or the B-mode ultrasonography. Only 3.78% received hormone replacement therapy treatment; the others were prescribed the traditional Chinese medicine, supplements of calcium, or prescription drugs for heart disease, etc. (Table 3).

### 3.4. Factors Associated with Healthcare-Seeking Behaviors

The chi-squared test showed that age, employment status, health status of the husband, menopause status, KMI, and the number of symptoms had significant association with the healthcare-seeking behavior (*p* < 0.01) (Table 4).

The logistic model analysis revealed that employment status, menopause status, and KMI groups were significantly associated with healthcare-seeking behaviors. Compared with the “not obvious” group, the severe group of the KMI (odds ratio (OR) = 4.274, 95% confidence interval (CI): 1.732–2.985) and the moderate group of the KMI (OR = 4.654, 95% CI: 2.389–9.067) had more possibilities to go to clinics. Compared with the women in the premenopausal period, the women in the perimenopausal and postmenopausal periods had more possibilities to seek for healthcare, with OR 1.621 (95% CI: 1.165–2.256) and OR 1.356 (95% CI: 1.010–1.821), respectively. Unemployed women (OR = 1.445, 95% CI: 1.133–1.842) were more likely to go to clinics for healthcare (Table 5).

## 4. Discussion

We investigated prevalence of menopausal symptoms in the community-based population of 40–60-years-old women and their healthcare-seeking behaviors in Shanghai, China. This is one of few research studies in China indicating prevalence of menopausal symptoms in middle-aged women with a relatively large sample size. However, as Shanghai is one of the most developed cities in China, the results might not be generalized to the Chinese women in other provinces.

Among the 3147 respondents included in the analysis, 14.52% were perimenopausal women. The mean age of the menopause was 49.89 years, and 80% of the respondents had the menopausal age in the range of 46–54 years. The age range of the menopause in Shanghai women was similar to the women in other cities in China [15,16,17], but lower than in the women in Western countries, with an age range of 50–52 years.

The results showed more severe symptoms during the perimenopausal period comparing to the premenopausal period. The prevalence of hot flushes/sweating, fatigue, and emotional disorder among the participants in the perimenopausal period was significantly higher than in the women in the pre- and postmenopausal period. A longitudinal study of 2565 women in Massachusetts showed an increase in symptom reporting rates during the perimenopausal period, especially immediately prior to the menopause [18]. As suggested by researchers, major physiological changes due to the hormone level decline may contribute to the increase in symptom complaints during this period. The level of symptom reporting was different by race/ethnicity, which is why there was a difference between American and Asian women [19]. However, a multisite study of women’s health across the nation (SWAN) showed that ethnicity was no longer significant when analyses were adjusted for socioeconomic status, health, lifestyle, or social circumstances [20].

In our study, the prevalence of menopausal symptoms in the age range of 40–60 years was 73.8%, lower than in other countries [6]. In Germany, 81% of the investigated population had at least one menopausal complaint [21]. In our study, the most common symptom of the participants was fatigue, which took up 38.08% of the total, and the second was hot flushes/sweating, which accounted for 33.65%. These reported symptoms differed from the Western countries. In the Nancy Fugate Woods’ review, hot flushes were the predominant symptom with a prevalence rate of up to 33–63% during a late menopausal transition [22]. Vasomotor symptoms had the highest prevalence rates among German women (71.2%). However, in Asian women, such as Indian and Japanese women, the prevalence of vasomotor symptoms was much lower than in the Western countries [23,24]. Similar results were seen in Chinese women, where vasomotor symptoms were also not in the top ranking. In a community-based study, fatigue was the most prevalent symptom, the same finding as in our study [25]. Muscle/joint pain and sexual problems were also very common in Chinese women [15]. In recent years, emotional disorders related to the menopause have drawn attention, the prevalence of it in our study was 21.25% during the perimenopausal period, which was lower than in some reported studies in the Western countries, which ranged from 23% to 34% [26,27].

There were several ways of arresting menopausal symptoms as reported, such as physical activity, alternative medicine, and a healthy diet [28]. In the beginning of the 21st century, alternative approaches to the menopause have frequently been discussed, such as alteration of the lifestyle, black cohosh, yoga, homeopathy, traditional Chinese medicine (TCM), etc., and showed some effects [21,29]. However, women with severe and multiple symptoms need medical support. Medical support can alleviate severity of menopausal symptoms and shorten their duration. In our study, only 36.77% of the women who had menopausal symptoms went to see a doctor. If they did seek help at a hospital, the most common clinic they visited was the internal medicine department (39.43% of the total number of help-seeking women). The second most common clinic they went to visit is the gynecology department (22.17%). The number of women visiting the endocrinology and menopausal departments only accounted for 9.98% and 8.49%, respectively. The results implied that most of the patients did not know which clinics they should visit when they had menopause symptoms, or they did not even know the symptoms had a relationship with their hormonal changes.

The menopausal hormone therapy (MHT) was the most effective therapy for women suffering from the menopausal syndrome; the International Menopause Society (IMS) provided evidence-based recommendations on the use of the MHT in 2013, and updated details in 2016. Evidence shows that, if used properly, it can reduce the risk of some chronic diseases, such as osteoporosis [3]. In the past decades, menopausal hormone therapy (MHT) has been used as one of the most broadly prescribed therapies in the Western countries [30]. Despite its wide use in the West, in China, women knew little about MHT’s benefits and risks; consequently, they were less likely to choose this treatment. A study showed that less than 1% of Chinese women suffering from menopausal symptoms used the MHT [31]. In our study, women receiving MHT were also very rare, the percentage was only 3.78%. The most common treatments were the traditional Chinese medicine and other drugs, which took up 43.58% and 20.91%, respectively. It revealed that doctors, even gynecologists, did not prescribe the MHT to women who had menopausal symptoms, indicating that they also lacked knowledge regarding the menopause and the use of the MHT.

The results of the logistic regression analysis revealed that the age period, employment status, menstruation status, and KMI groups remained influencing factors relevant to whether women sought healthcare services. More unemployed and retired women than employed ones tended to go to hospital. This might be related to much more leisure time to care for their health and visits to hospitals. Most of the retired women in Shanghai had medical insurance, which might be another reason.

The number of symptoms was found to be a major factor influencing the women’s healthcare-seeking behavior in our study. The women with more than four symptoms were more likely to visit hospitals, compared to those women suffering from three or less symptoms. This result was similar to a review, in which women would not seek treatment unless they experienced multiple or severe symptoms [32]. It was also consistent with one of our other hospital-based studies, which showed women without symptoms or with only two or fewer menopausal symptoms were less likely to visit hospitals promptly compared to those experiencing five or more symptoms [33].

Severity of menopausal symptoms was also a major factor influencing the women’s healthcare-seeking behavior. The women with moderate and severe symptoms were more likely to visit hospitals compared to the women with mild or no symptoms. Similar results were found by other researchers [33].

In our study, most socioeconomic characteristics had no relationship with women’s healthcare-seeking behaviors, except for the husband’s health status. Those women whose husbands had diseases tended to be more likely to seek care at clinics for their menopausal symptoms. We did not analyze the influencing factors of women’s menopausal symptoms, but in some studies, factors including lifestyle, chronic disease, anxiety, and depression were found to be associated with self-perceived menopausal symptoms in Chinese women [15]. The husband’s disease might cause high pressure and anxiety to women and intensify their menopausal symptoms. This might increase their awareness of their own health and make them see a doctor timely.

## 5. Conclusions

The prevalence of menopausal symptoms was relatively high among middle-aged Shanghai women, and the perimenopausal women had the highest symptom level. However, only a small percentage of the participants sought healthcare because of menopausal symptoms. Most of them visited the department of internal medicine and the gynecology department. Older women, unemployed or retired women, women with more menopausal symptoms and more severe symptoms were more likely to seek healthcare services. The menopause is a natural physiological transition in a woman’s life, but it is essential for women to be aware of the adverse aspects of the menopause and to be prompted to seek prevention and treatment for their health [17]. Health education to improve women’s knowledge of the menopause and change their attitude towards menopausal healthcare is necessary.

## 6. Limitation of the Study

Though the sample size was relatively large compared to other studies in China, we used stratified and quota sampling, and the districts studied were not randomly selected, so the sample could not represent the total population in Shanghai. When we recruited subjects of the study, we did not exclude women with the irregular menstrual period or those who skipped the perimenopausal stages. This might cause some bias of the study.

## Figures and Tables

**Table 1 ijerph-17-02640-t001:** General socioeconomic and health characteristics of the study population (*n* = 3147).

Characteristics	*n*	%
Age (mean ± SD)	51.2 ± 5.80	
40–44	556	17.67
45–49	568	18.05
50–54	841	26.72
55–59	1182	37.56
Education		
Bachelor’s degree or higher	348	11.06
College	471	14.97
High school	1313	41.72
Junior middle school or lower	1015	32.25
Employment status ^a^		
Employed	1524	48.81
Unemployed/retired	1598	51.19
Marital status ^a^		
Married	2813	91.54
Divorced, unmarried, widowed	260	8.46
Average household income (CNY/month) ^a^		
≥10,000	442	14.19
5000–9999	772	24.79
3000–4999	1053	33.82
≤2999	847	27.20
Number of family members living together ^a^		
1	51	1.63
2	712	22.73
3	1543	49.27
≥4	826	26.37
Health insurance ^a^		
Has insurance	3035	97.12
No insurance	90	2.88
Menopause status ^a^		
Pre-	1036	33.13
Peri-	454	14.52
Post-	1638	52.35
Health status of the husband ^b^		
Healthy	2378	84.48
Has a disease	435	15.46

^a^ The number of missing values—employment status (25); marital status (74); average household income (CNY/month) (33); number of family members living together (15); health insurance (22); menopause status (19); ^b^
*n* = 2813. SD—standard deviation; CNY—Chinese Yuan.

**Table 2 ijerph-17-02640-t002:** Menopausal symptoms and the mKMI of the participants.

Variables	Premenopausal Stage (*n*_1_ = 1020) *n* (%)	Peri-Menopausal Stage (*n*_2_ = 447) *n* (%)	Postmenopausal Stage (*n*_3_ = 1612) *n* (%)	*p*	Total
Menopausal symptoms ^a^	641 (62.8)	366 (81.9)	1268 (78.7)	<0.001	2318 (73.8)
Hot flushes and sweating	169 (16.57)	193 (43.18)	679 (42.12)	<0.001	1057 (33.65)
Sleep disturbance	197 (19.31)	135 (30.20)	517 (32.07)	<0.001	863 (27.48)
Rapid heartbeat	125 (12.25)	78 (17.45)	284 (17.62)	0.001	499 (15.89)
Emotional disorder	158 (15.49)	95 (21.25)	308 (19.11)	0.013	574 (18.27)
Sexual dysfunction	10 (0.98)	9 (2.01)	69 (4.28)	<0.001	91 (2.90)
Dysuria or frequent urination	17 (1.67)	12 (2.68)	55 (3.41)	0.028	86 (2.74)
Uracratia	7 (0.69)	17 (3.80)	53 (3.29)	<0.001	79 (2.52)
Joint ache	197 (19.31)	147 (32.89)	545 (33.81)	<0.001	905 (28.81)
Height decrease	38 (3.73)	22 (4.92)	144 (8.93)	<0.001	210 (6.69)
Bone fracture	14 (1.37)	8 (1.79)	49 (3.04)	0.016	72 (2.29)
Fatigue	417 (40.88)	199 (44.52)	555 (34.43)	<0.001	1196 (38.08)
mKMI score ^b^					
0–6	620 (63.01)	181 (41.90)	627 (41.06)	<0.01	1542 (51.49)
7–15	277 (28.15)	157 (36.34)	559 (36.61)		1012 (33.79)
16–30	80 (8.13)	90 (20.83)	317 (20.76)		419 (13.99)
≥31	7 (0.71)	4 (0.93)	24 (1.57)		22 (7.35)

^a^ missing value = 6; ^b^ missing value = 152. mKMI—modified Kupperman menopausal index.

**Table 3 ijerph-17-02640-t003:** Experiences in healthcare seeking for women’s menopausal complaints (*n* = 602).

Variables	*n*	%
Clinic departments		
Internal medicine department	237	39.43
Gynecology department	133	22.17
Endocrinology department	60	9.98
Menopausal department	51	8.49
Urology department	27	4.49
Other	258	42.93
Check-ups		
General blood test	303	50.67
Electrocardiogram (ECG)	259	43.31
B-mode ultrasonography	258	43.14
Computed tomography (CT)	173	28.93
Bone density	152	25.42
Estrogens	63	10.54
Others	51	8.53
Perimenopausal scales	24	4.01
No check-ups	78	13.04
Diagnosis		
No definite diagnosis	246	42.12
Perimenopausal symptoms	96	16.44
Perimenopausal syndrome	71	12.16
Other	171	29.28
Treatment		
Traditional Chinese medicine	173	43.58
Other drugs	83	20.91
Supplements of calcium	71	17.88
other	67	16.88
Drugs for heart disease	57	14.36
Sleeping pills	42	10.58
Menopausal hormone therapy (MHT)	15	3.78
Mental consultation	8	2.02
Pelvic surgery	3	0.76

**Table 4 ijerph-17-02640-t004:** Influencing factors of seeking healthcare among women with menopausal symptoms.

Variables	Yes *n* (%)	No *n* (%)	*p*
Number of participants	602	1637	
Age, years			
40–44	64 (18.60)	280 (81.40)	<0.001
45–49	83 (22.07)	293 (77.93)	
50–54	178 (28.12)	453 (71.88)	
55–59	277 (31.26)	608 (68.74)	
Education			
Bachelor’s degree or higher	52 (21.67)	189 (78.33)	0.082
College	82 (24.77)	246 (75.23)	
High school	254 (27.05)	688 (72.95)	
Junior middle school or lower	214 (29.68)	507 (70.32)	
Employment			
Employed	230 (21.99)	816 (78.01)	<0.001
Unemployed	372 (31.18)	821 (68.82)	
Marital status			
Married	541 (27.10)	1455 (72.90)	0.534
Other	47 (25.00)	141 (75.00)	
Average household income (CNY/month)			
≥10,000	78 (25.66)	226 (74.34)	0.744
5000–9999	142 (26.06)	403 (73.94)	
3000–4999	196 (27.11)	527 (72.89)	
≤2999	182 (28.48)	457 (71.52)	
Number of family members living together			
1	7 (19.44)	29 (80.56)	0.034
2	163 (31.17)	360 (68.83)	
3	263 (24.72)	801 (75.28)	
≥4	168 (27.81)	436 (72.19)	
Health insurance			
Has insurance	579 (26.83)	1579 (73.17)	0.454
No insurance	19 (31.15)	42 (68.85)	
Menopause status			
Pre-	110 (17.89)	505 (82.11)	<0.001
Peri-	104 (29.46)	249 (70.54)	
Post-	372 (30.37)	853 (69.63)	
KMI			
Normal	143 (20.40)	558 (79.60)	<0.001
Slight	218 (24.22)	682 (75.78)	
Moderate	204 (39.46)	313 (60.54)	
Severe	11 (52.38)	10 (47.62)	
Number of symptoms			
1–3	421 (23.32)	1384 (76.68)	<0.001
>3	181 (41.80)	252 (58.20)	
Health status of husband			
Healthy	426 (25.76)	1228 (74.24)	<0.001
Has disease	123 (35.45)	224 (64.55)	
Total	471 (29.64)	1118 (70.36)	

KMI—Kupperman menopausal index.

**Table 5 ijerph-17-02640-t005:** The logistic analysis for the factors associated with healthcare seeking among women with menopausal symptoms.

Independent Variables	Comparative Group	Control Group	OR (95% CI)	*p*
Employment	Unemployment	Employment	1.445 (1.133–1.842)	0.003
Number of symptoms	More than 4 symptoms	0–3 symptoms	2.12 (1.493–3.021)	<0.001
Menopause status	Peri-	Pre-	1.621 (1.165–2.256)	0.018
	Post-		1.356 (1.010–1.821)	0.611
KMI group	Mild	Normal	1.140 (0.889–1.462)	0.253
	Moderate		3.041 (1.211–7.632)	0.018
	Severe		4.654 (2.389–9.067)	0.004
Health status of the husband	Unhealthy	Healthy	1.403 (1.078–1.828)	0.012

CI—confidence interval; OR—odds ratio.

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
