# Peer review of "Menopausal Symptoms and Perimenopausal Healthcare-Seeking Behavior in Women Aged 40–60 Years: A Community-Based Cross-Sectional Survey in Shanghai, China"

_ijerph, 2020, doi:10.3390/ijerph17082640_

Round 1

Reviewer 1 Report

Major Comment for Authors:

It is an excellent idea to assess the midlife experiences of Shanghai women, and to assess whether they seek health care assistance. However, the wide age range (40-60 years) spans three distinct reproductive life phases—premenopause, perimenopause and postmenopause. Each has different hormonal levels and different experiences. The only ones peri- and postmenopausal women share are hot flushes/night sweats and vaginal dryness/lower interest in sex.  In addition, the Kupperman Index is validated only for postmenopausal women.

The language and approach of this paper implies that women “should be” symptomatic and “should” use some kind of therapy.  

Specific Necessary Changes:

  1. Please describe each symptom by its experience in women of a different reproductive phase—e.g. vasomotor symptoms in perimenopausal women.
  2. Please describe the sampling so it will be clear whether it is a random sample or a convenience one—and state that in the abstract and methods.
  3. In the background please quote the currently accepted way of designating women as pre/peri/postmenopausal—Stages of Reproductive Aging Workshop 10+1.
  4. Please also, in the methods, describe the overall hormonal differences between women who are premenopausal, perimenopausal2 and postmenopausal.  
  5. Provide a table showing demographic data for Shanghai women as a whole versus those in the study (as Ho3 did) so the reader can tell how representative these surveyed women are.
  6. How did you deal with women whose cycles were not regular or who had skipped fewer than three cycles—did you call them premenopausal, did you exclude them?
  7. Please have a fluent or native English speaker/reader review and correct language issues.

Reference List

  1. Harlow SD, Gass M, Hall JE, et al. Executive summary of the Stages of Reproductive Aging Workshop +10: addressing the unfinished agenda of staging reproductive aging. Climacteric 2012; 15(2): 105-14.
  2. Prior JC. Perimenopause: The complex endocrinology of the menopausal transition. Endocrine Reviews 1998; 19: 397-428.
  3. Ho SC, Chan SG, Yip YB, Cheng A, Yi Q, Chan C. Menopausal symptoms and symptoms clustering in Chinese women. Maturitas 1999; 33: 219-27.

Author Response

Specific Necessary Changes:

  1. Please describe each symptom by its experience in women of a different reproductive phase—e.g. vasomotor symptoms in perimenopausal women.
  2. Please describe the sampling so it will be clear whether it is a random sample or a convenience one—and state that in the abstract and methods.
  3. In the background please quote the currently accepted way of designating women as pre/peri/postmenopausal—Stages of Reproductive Aging Workshop 10+1.
  4. Please also, in the methods, describe the overall hormonal differences between women who are premenopausal, perimenopausal2 and postmenopausal.  
  5. Provide a table showing demographic data for Shanghai women as a whole versus those in the study (as Ho3 did) so the reader can tell how representative these surveyed women are.
  6. How did you deal with women whose cycles were not regular or who had skipped fewer than three cycles—did you call them premenopausal, did you exclude them?
  7. Please have a fluent or native English speaker/reader review and correct language issues.

Answer to point 1:

We have revised the paragraph which described the symptoms in different phase.

Answer to point 2:

The sampling is not random and described in the method.

Answer to point 3:

 In the background please quote the currently accepted way of designating women as pre/peri/postmenopausal—Stages of Reproductive Aging Workshop 10+1

Answer to point 4: We have describe the overall hormonal differences between women who are premenopausal, perimenopausal and postmenopausal.

Answer to point 5: The statistic system is different between in Shanghai and in Hongkong. We cann't find the overall demographic data listed in the article. The sampling procedure showed that it was not a random sample, so it couldn't represent the whole population in Shanghai.

Answer to point 6: In the design of the study, we didn't exclude women with irregular cycles or skipped fewer than three cycles. We will write it in the limitation of the study.

Answer to point 7: We have asked a native English speaker to revise the language.

Reviewer 2 Report

The manuscript still needs extensive editing of English language/style.

The title of the manuscript is misleading because it talks about “Health Care Seeking”. The title should be adapted accordingly.

Keywords. China isn't a MeSH term. 

We recommend updating the data to 2019. [64] In China, the number of female population at 40-60 years was about 322 million in 2015 with a life expectance at 79.43 years in female population.

2.1 Study design and sampling. It is necessary to put Exclusion criteria.

It is necessary to review the capital words in the text [99-102] and spaces between words.

[134] and jointache were the most common symptoms, accounting for 38.08%, 33.65%and 28.81%.

[149] There were 602participants see

You must review all references, specially: 1, 2, 3, 4, 9, 11, 12, 13, 15, 16, 17, 20, 22, 30, 38. Author 1, A.B.; Author 2, C.D. Title of the article. Abbreviated Journal Name YearVolume, page range.

Author Response

Comments and Suggestions for Authors

point1: The manuscript still needs extensive editing of English language/style.

Reply: We have asked a native English speaker to revise the language for us.

Point 2: The title of the manuscript is misleading because it talks about “Health Care Seeking”. The title should be adapted accordingly.

Reply: The title of the manuscript has been changed to: Menopausal Symptoms and Perimenopausal Healthcare Seeking Behavior in Women Aged 40-60 Years: A Community-based Cross-sectional Survey in Shanghai, China

Point 3: Keywords. China isn't a MeSH term. 

Reply: China has been deleted as MeSH term.

Point4: We recommend updating the data to 2019. [64] In China, the number of female population at 40-60 years was about 322 million in 2015 with a life expectance at 79.43 years in female population.

Reply: We searched government's website about the data again, and couldn't find the specific numbers between 40-60 years after 2015. The data was calculated in different way in recent years.

Point 5: 2.1 Study design and sampling. It is necessary to put Exclusion criteria.

Reply: We have put Exclusion criteria in the method.

Point 6: It is necessary to review the capital words in the text [99-102] and spaces between words.

Reply: We have revised the capital words in the text [99-102]and spaces between words

Point 7: [134] and jointache were the most common symptoms, accounting for 38.08%, 33.65%and 28.81%.

Reply: We have revised the space between words.

[149] There were 602participants see

Reply: We have revised the space between words.

You must review all references, specially: 1, 2, 3, 4, 9, 11, 12, 13, 15, 16, 17, 20, 22, 30, 38. Author 1, A.B.; Author 2, C.D. Title of the article. Abbreviated Journal Name YearVolume, page range.

Reply: We have checked the references and did revision.

Reviewer 3 Report

The study aims to evaluate the prevalence and severity of menopausal symptoms among middle-aged women and to understand the factors associated with women’s perimenopausal health care seeking behavior in Shanghai, China. A community-based cross-sectional study was carried out involving 3147 participants aged 40-60 years in 7 districts. The authors found the total prevalence of menopausal symptoms was 73.8%, while among perimenopausal women it was the highest (81.70%). Perimenopausal and postmenopausal participants had higher symptoms scored with mKMI than premenopausal women (p<0.01). Of the total women who had symptoms, 25.97% had sought health care. Logistic regression model revealed that employment, menstruation status and mKMI were significantly associated with healthcare seeking behaviors (p<0.01). Therefore, they concluded that the prevalence of menopausal symptoms was relatively high among middle-aged women, and perimenopausal women had the highest level. However, only a small percentage of the participants sought health care. Carrying out health education may be measures to improve health seeking behavior. However, this study is merely a comparison of symptoms and medication among different age groups and the health-care-seeking behavior without detailed and in-depth discussion. Moreover, there is a bottom of articles describing such differences. Therefore the manuscript is a bit less creative and may add not much to current knowledge. A major revision is needed for the manuscript before its publication.

My comments are as follows:

  1. As mentioned above, there is a bottom of articles describing such differences for menopausal women. In the section of discussion, the authors should add detailed and in-depth discussion for their observations, as well as the possible interventions. The authors may consider adding some descriptions for discusson since they have reported the results.
  2. The participants were sampled with a method of purposive sampling for the case groups, which may confound this result of the study. The districts (urban, rural…), the reasons and process of purposive sampling should be detailed further in the Section of Methods. Otherwise this should be put into the limitation of the study.
  3. The description of percentages in the first two rows (sweating, sleep disturbance) of Table 2 seems wrong since they are completely the same in the first two rows. Please clarify.
  4. The authors mentioned that “There were 602 participants seeking health care services, accounting for 25.97% of the total women with menopausal symptoms. Only 8.49% of them went to menopausal department, majority of them went to internal department (39.43%) and gynecologic department (22.17%). Only 10.54% had estrogen test and 4.01% had perimenopausal sacle test, most of them had general blood test and ECG B-mode ultrasonography test. (Line 149-153)” However, the reasons for seeking health care services were multiple, not only menopausal symptoms but also other medical etiologies. Please clarify.
  5. The sentences in Line 229-231 “The results of the logistic analysis revealed that age was not a significant factor related to women's health care seeking behavior. This may because age had strong relationship with other factors, such as menopause status.” is not reasonable, and is inconsistent with the result (p<0.001) of the first row in Table 4. Please remove the word “not” in the sentences mentioned above.
  6. The style of references is incorrect. Please make changes for your references to conform to the journal’s style.

Author Response

Comments of the reviewer:

  1. As mentioned above, there is a bottom of articles describing such differences for menopausal women. In the section of discussion, the authors should add detailed and in-depth discussion for their observations, as well as the possible interventions. The authors may consider adding some descriptions for discusson since they have reported the results.
  2. The participants were sampled with a method of purposive sampling for the case groups, which may confound this result of the study. The districts (urban, rural…), the reasons and process of purposive sampling should be detailed further in the Section of Methods. Otherwise this should be put into the limitation of the study.
  3. The description of percentages in the first two rows (sweating, sleep disturbance) of Table 2 seems wrong since they are completely the same in the first two rows. Please clarify.
  4. The authors mentioned that “There were 602 participants seeking health care services, accounting for 25.97% of the total women with menopausal symptoms. Only 8.49% of them went to menopausal department, majority of them went to internal department (39.43%) and gynecologic department (22.17%). Only 10.54% had estrogen test and 4.01% had perimenopausal sacle test, most of them had general blood test and ECG B-mode ultrasonography test. (Line 149-153)” However, the reasons for seeking health care services were multiple, not only menopausal symptoms but also other medical etiologies. Please clarify.
  5. The sentences in Line 229-231 “The results of the logistic analysis revealed that age was not a significant factor related to women's health care seeking behavior. This may because age had strong relationship with other factors, such as menopause status.” is not reasonable, and is inconsistent with the result (p<0.001) of the first row in Table 4. Please remove the word “not” in the sentences mentioned above.
  6. The style of references is incorrect. Please make changes for your references to conform to the journal’s style.

Answers to comments:

Answer to comment 1: we did revision in the discussion section.

Answer to comment 2: The participants were sampled with a method of purposive sampling for the case groups, so it's not a random sample. We will write in the limitation of the research.

Answer to comment 3: The description of percentages in the first two rows (sweating, sleep disturbance) of Table 2 was wrong , we have corrected it.

Answer to comment 4: Yes, the reasons for seeking health care services were multiple, not only menopausal symptoms. So in the questionnaire, we listed all of the symptoms of menopause and asked them whether they go to see the doctor just because of these symptoms.

Answer to comment 5: Thank you for the reminder. We deleted this sentence in the paragraph.

Answer to comment 6: We made the changes for the references.

Round 2

Reviewer 3 Report

.

Author Response

We further revised the limitation of the study, and the conclusion part of the article.